

# Retrospective study: anthropometric and metabolic characteristics of patients with metabolic syndrome–a gender-specific analysis of clinical and correlation patterns

Yingxin Li[1,*], Huanhua Wu[2,*], Yingtong Lu[1], Jingjie Shang[1], Yong Cheng[1], Xiaozheng Cao[2], Zhiqiang Tan[1], Qijun Cai[1], Hao Xu[1] and Bin Guo[1]

[1] Department of Nuclear Medicine and PET/CT-MRI Center, The First Affiliated Hospital of Jinan University, Guangzhou, Guangdong, China
[2] Central Laboratory, The Affiliated Shunde Hospital of Jinan University, Guangzhou, Guangdong, China
* These authors contributed equally to this work.

Corresponding author
Bin Guo, hyxguobin@126.com

## ABSTRACT

**Background:** Metabolic syndrome (MetS) is a prevalent condition associated with increased cardiometabolic risk. Despite its clinical significance, the relationships between anthropometric indices and metabolic markers—particularly in a gender-specific context—remain inadequately explored.

**Methods:** This retrospective study included 104 patients diagnosed with MetS between 2016 and 2022. Anthropometric indices, including body roundness index (BRI), body mass index (BMI), waist-to-height ratio (WHtR), and weight-to-waist index (WWI), were analyzed for their associations with metabolic markers reflecting insulin resistance and glucose metabolism: fasting blood glucose (FBG), fasting insulin (FIns), and homeostatic model assessment of insulin resistance (HOMA-IR). These markers represent key components of MetS, although other diagnostic elements such as dyslipidemia and hypertension were beyond the scope of this analysis. Gender-specific subgroup analyses were conducted for 57 female and 47 male patients.

**Results:** Significant differences in anthropometric and metabolic characteristics were observed between genders. Male patients exhibited higher BMI (40.4 [37.5, 44.2] *vs.* 37.2 [33.6, 40.3], $p = 0.022$), waist circumference (128.0 [120.2, 137.5] cm *vs.* 115.0 [106.0, 120.0] cm, $p < 0.001$), and FIns levels (22.8 [16.4, 29.5] μIU/mL *vs.* 17.8 [15.0, 23.4] μIU/mL, $p = 0.107$). Correlation analyses revealed that BMI and BRI were strongly associated with HOMA-IR and FIns in both genders, with stronger associations observed in males (*e.g.*, BMI *vs.* FIns, r = 0.60, $p < 0.001$). WHtR was significantly correlated with metabolic markers in males but not in females. WWI showed limited correlations across both genders.

**Discussion:** This study highlights distinct gender-specific patterns in the relationships between anthropometric indices and metabolic markers in MetS patients. These findings underscore the importance of tailored strategies in managing MetS, particularly considering gender-based differences in clinical and metabolic profiles.

## STUDY IMPORTANCE

What is already known?

- Indices such as body mass index (BMI), waist-to-height ration (WHtR), and body roundness index (BRI) are widely recognized as reliable markers for evaluating central obesity and its association with metabolic dysfunction. Their utility in predicting insulin resistance and related metabolic risks has been extensively validated in various populations.
- Men typically exhibit greater visceral fat accumulation, while women tend to have higher subcutaneous fat deposits. These gender-specific patterns influence the strength of correlations between anthropometric indices and metabolic markers, with stronger associations generally observed in males.

What does this study add?

- This study reveals stronger correlations between anthropometric indices (e.g., BMI, BRI, WHtR) and metabolic markers in males compared to females, highlighting the necessity of incorporating gender-specific assessments in the evaluation and management of metabolic syndrome (MetS).
- The study identifies BRI as a robust predictor of metabolic dysfunction, with comparable or superior performance to traditional indices like BMI and WHtR, particularly in males. This suggests BRI's potential as a valuable alternative marker for central obesity and insulin resistance.
- The findings demonstrate limited predictive value of weight-to-waist index (WWI) for metabolic dysfunction across genders, challenging its clinical utility and advocating for the development of more reliable metrics tailored to diverse populations.

How might these results change the direction of research or the focus of clinical practice?

- These results emphasize the need for gender-specific assessments and personalized approaches in evaluating and managing MetS, while highlighting the potential of alternative anthropometric indices like the BRI to refine risk stratification and early intervention strategies in clinical practice.

## INTRODUCTION

MetS is a multifaceted disorder characterized by a cluster of metabolic abnormalities, including central obesity, insulin resistance (IR), dyslipidemia, and elevated blood pressure (*Lemieux & Després, 2020*; *Neeland et al., 2024*). It is associated with an increased risk of cardiovascular diseases and type 2 diabetes, rendering it a critical public health concern globally (*Noubiap et al., 2022*). Anthropometric indices, such as BMI, BRI

(*Zhang et al., 2024b*), WHtR (*Cameron, Magliano & Söderberg, 2013*), and WWI (*Park et al., 2018*), are widely employed for assessing obesity and its associated risks. These indices provide valuable insights into fat distribution and metabolic dysfunction (*Huang et al., 2022*; *Gebremedhin & Bekele, 2024*). In addition to anthropometric indices, bioelectrical impedance analysis can serve as a valuable complementary tool, offering precise estimates of fat mass, lean mass, and total body water, thereby enhancing the assessment of metabolic risk (*Son et al., 2025*).

Insulin resistance plays a pivotal role in the pathophysiology of MetS (*Muzurović, Mikhailidis & Mantzoros, 2021*). It is closely linked to obesity-related metabolic disturbances and is exacerbated by visceral and ectopic fat accumulation. Recent global studies have demonstrated substantial variability in anthropometric and metabolic characteristics across different populations (*Bajaj et al., 2025*). For example, Western populations tend to exhibit higher absolute BMI and waist circumference values compared to East Asian cohorts, even at similar levels of metabolic risk, highlighting ethnic differences in visceral adiposity and insulin sensitivity (*Iliodromiti et al., 2023*; *Eng et al., 2025*).

Sex-specific differences in metabolic disease progression are also well recognized (*Lin et al., 2021*). Women are more likely to develop subcutaneous fat accumulation, whereas men are prone to visceral and ectopic fat deposition, which accelerates insulin resistance and cardiometabolic complications (*Gavin & Bessesen, 2020*; *Karastergiou et al., 2012*). Moreover, hormonal influences, such as estrogen's protective effect on lipid metabolism, partially account for delayed MetS onset in premenopausal women, while postmenopausal changes reverse this advantage. These physiological differences result in distinct correlations between anthropometric indices and metabolic markers in males *vs*. females, as shown in multi-ethnic studies (*Carter et al., 2023*). Despite extensive research, there remains a lack of comprehensive studies investigating the correlations between anthropometric indices and metabolic markers in a gender-specific context. Such analyses are crucial for understanding the nuanced interplay between obesity and metabolic dysfunction and for tailoring preventive and therapeutic strategies. Addressing this gap is critical for tailoring preventive strategies and refining risk stratification models that account for both ethnic and sex-specific differences.

Therefore, this retrospective study aimed to (1) characterize gender-specific anthropometric indices—including BRI, BMI, WHtR, and WWI—and metabolic markers, such as fasting blood glucose (FBG), fasting insulin (FIns), and homeostatic model assessment of insulin resistance (HOMA-IR), in patients with MetS; and (2) examine gender-specific correlations between these indices and metabolic markers to provide empirical evidence for personalized clinical management. Subgroup analyses were also conducted to explore the potential clinical implications of these gender-related associations. Although MetS encompasses multiple interrelated components, including dyslipidemia and hypertension, this study focused primarily on the relationship between central obesity-related anthropometric measures and insulin resistance markers, given their central role in the pathophysiology of MetS.

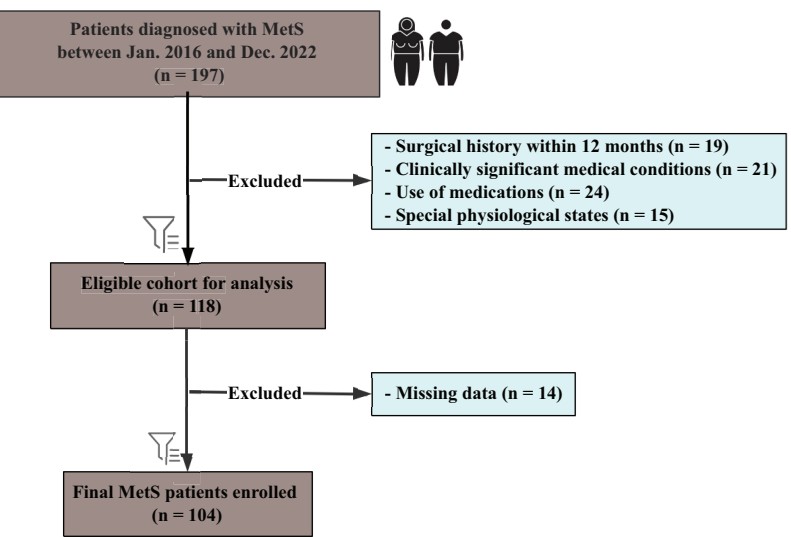

**Figure 1 Flowchart of patient selection for the study cohort.** Flowchart illustrating the selection process for patients diagnosed with metabolic syndrome (MetS) between January 2016 and December 2022.

## MATERIALS AND METHODS

### Study participants

As shown in Fig. 1, this retrospective study included patients diagnosed with MetS between January 2016 and December 2022, following the International Diabetes Federation guidelines. Participants aged 18–60 years were eligible if they met at least three of the following criteria: central obesity (waist circumference ≥90 cm in men or ≥80 cm in women, Asian standards), triglycerides ≥1.7 mmol/L, reduced high-density lipoprotein cholesterol (HDL-C) (<1.03 mmol/L in men or <1.29 mmol/L in women), blood pressure ≥130/85 mmHg or receiving antihypertensive treatment, or FBG ≥5.6 mmol/L or a confirmed type 2 diabetes diagnosis. All enrolled patients fulfilled at least three of these criteria; however, the specific combination of MetS components varied among individuals rather than being identical across the cohort. Patients with previously diagnosed type 2 diabetes or those receiving stable glucose-lowering therapy were also considered to meet the hyperglycemia criterion. Exclusion criteria included: a history of bariatric or other major surgical interventions within the preceding 12 months ($n = 19$); clinically significant medical conditions ($n = 21$), defined as active malignancies, severe cardiovascular diseases, severe renal or hepatic impairment, uncontrolled psychiatric disorders, or immunodeficiency conditions; use of medications known to affect metabolic function or body weight ($n = 24$); and special physiological states such as pregnancy or lactation ($n = 15$).

From 197 initially identified patients, 118 were eligible after exclusions, with a final cohort of 104 patients (57 female, 47 male) after removing 14 cases due to missing data. Subgroup analyses were performed based on gender. This study was approved by the institutional review board of the First Affiliated Hospital of Jinan University

(Approval No. KY-2024-184) and conducted in accordance with national legislation and the Declaration of Helsinki. The need for informed consent was waived due to the retrospective design, and all data were anonymized.

### Anthropometrics and laboratory assessments

Anthropometric measurements were performed by trained healthcare professionals following standardized protocols. Height and weight were measured with participants wearing light clothing and no shoes, and BMI was calculated as weight (kg) divided by height squared (m$^2$). Waist circumference was measured at the midpoint between the lower margin of the last palpable rib and the top of the iliac crest, with participants in a standing position. Additional indices, including the BRI, WHtR, and WWI, were derived using established formulas to assess central obesity and fat distribution.

Laboratory assessments included FBG, triglycerides (TG), and FIns. Blood samples were collected after an overnight fast of at least 8 h and analyzed using standardized enzymatic methods (*Mei et al., 2024*). Insulin resistance was evaluated using the HOMA-IR, calculated as [fasting insulin (μIU/mL) × fasting glucose (mmol/L)] ÷ 22.5. Quality control measures were implemented throughout to ensure accuracy and reliability of all laboratory results.

### Calculations of obesity-related indices

Obesity-related indices were calculated to provide a comprehensive assessment of body composition and fat distribution. The BMI was derived as weight (kg) divided by height squared (m$^2$). The BRI was calculated using the formula:

$$BRI = 364.2 - 365.5 \times \sqrt{1 - \left( \frac{waist\ circumference\ (cm)}{2\pi \times Height\ (cm)} \right)^2}. \tag{1}$$

WHtR was computed as waist circumference (cm) divided by height (cm), and WWI was calculated as weight (kg) divided by waist circumference squared (m$^2$).

These indices were selected for their established utility in predicting central obesity and metabolic risk. Each calculation was performed using standardized software to ensure consistency, and results were cross-checked for accuracy. The derived indices were subsequently used for correlation analyses with metabolic markers, such as fasting insulin, fasting glucose, and HOMA-IR, to explore their relationships with insulin resistance and metabolic dysfunction.

### Statistical analysis

All statistical analyses were performed using R software (version 4.3.1). The Shapiro-Wilk test was applied to assess the normality of continuous variables. Normally distributed variables were reported as mean ± standard deviation (SD) and analyzed using the independent Student's t-test, whereas non-normally distributed variables were expressed as median (interquartile range (IQR)) and evaluated with the Mann-Whitney U test. Categorical variables were summarized as frequencies (percentages) and compared using the Chi-square test. Correlations between anthropometric indices (BMI, BRI, WHtR,

**Table 1 Baseline clinical characteristics of 104 patients with MetS.**

| Variable[a] | Female (N = 57) | Male (N = 47) | p value[b] |
|---|---|---|---|
| Age, year, median (IQR) | 27.0 [24.0, 33.0] | 26.0 [22.0, 34.0] | 0.448 |
| Height, cm, (mean ± SD) | 163.3 ± 4.9 | 174.8 ± 5.9 | <0.001 |
| Weight, kg, (mean ± SD) | 97.1 ± 18.9 | 118.7 ± 21.1 | <0.001 |
| Waist, cm, (median [IQR]) | 115.0 [106.0, 120.0] | 128.0 [120.2, 137.5] | <0.001 |
| TG, mmol/L, (median [IQR]) | 1.5 [1.1, 2.1] | 1.8 [1.3, 2.6] | 0.131 |
| FBG, mmol/L, (median [IQR)) | 5.8 [4.9, 6.7] | 5.6 [5.0, 7.6] | 0.499 |
| FIns, µIU/mL, (median [IQR]) | 17.8 [15.0, 23.4] | 22.8 [16.4, 29.5] | 0.107 |
| HOMA_IR, (median [IQR)) | 4.6 [3.5, 7.9] | 6.4 [4.1, 9.5] | 0.061 |
| BRI, (median [IQR]) | 8.1 [6.8, 9.4] | 8.7 [7.2, 10.6) | 0.078 |
| BMI, (median [IQR)) | 37.2 [33.6, 40.3] | 40.4 [37.5, 44.2] | 0.022 |
| WWI, (mean ± SD) | 11.9 ± 1.2 | 11.9 ± 1.1 | 0.858 |
| WHtR, (median [IQR]) | 0.7 [0.7, 0.8] | 0.7 [0.7, 0.8] | 0.078 |

Notes:
[a] Continuous variables were presented as mean ± SD for normal distribution or median [IQR] for non-normal distribution and categorical variables as Number (%).
[b] p values were calculated using the Student's t-test for normally distributed or Wilcoxon Signed Rank test for non-normally distributed continuous variables and the $\chi^2$ test for categorical variables.
SD, standard error; IQR, interquartile range.

WWI) and metabolic markers (FBG, FIns, HOMA-IR) were analyzed using Pearson's correlation. Gender-specific subgroup analyses were conducted to account for potential confounders. Statistical significance was defined as $p < 0.05$, with results presented alongside 95% confidence intervals (CI).

## RESULTS

### Baseline characteristics

The baseline characteristics of the 104 patients with MetS included in the study are summarized in Table 1. Specifically, 29 participants (27.9%) had type 2 diabetes mellitus, 21 (20.2%) had hypertension, and 2 (1.9%) had gout. The cohort consisted of 57 females (54.8%) and 47 males (45.2%), with a median age of 27.0 years [24.0, 33.0] for females and 26.0 years [22.0, 34.0] for males ($p = 0.448$). Male patients had significantly higher mean height (174.8 ± 5.9 cm *vs.* 163.3 ± 4.9 cm, $p < 0.001$), weight (118.7 ± 21.1 kg *vs.* 97.1 ± 18.9 kg, $p < 0.001$), and waist circumference (128.0 [120.2, 137.5] cm *vs.* 115.0 [106.0, 120.0] cm, $p < 0.001$) compared to females.

FIns levels and HOMA-IR tended to be higher in males than females, although these differences did not reach statistical significance (FIns: 22.8 [16.4, 29.5] µIU/mL *vs.* 17.8 [15.0, 23.4] µIU/mL, $p = 0.107$; HOMA-IR: 6.4 [4.1, 9.5] *vs.* 4.6 [3.5, 7.9], $p = 0.061$). Obesity-related indices further highlighted gender-specific differences. Male patients exhibited significantly higher BMI (40.4 [37.5, 44.2] *vs.* 37.2 [33.6, 40.3], $p = 0.022$), while WWI and WHtR showed no significant differences between genders (WWI: 11.9 ± 1.1 *vs.* 11.9 ± 1.2, $p = 0.858$; WHtR: 0.7 [0.7, 0.8] *vs.* 0.7 [0.7, 0.8], $p = 0.078$).

Peer J

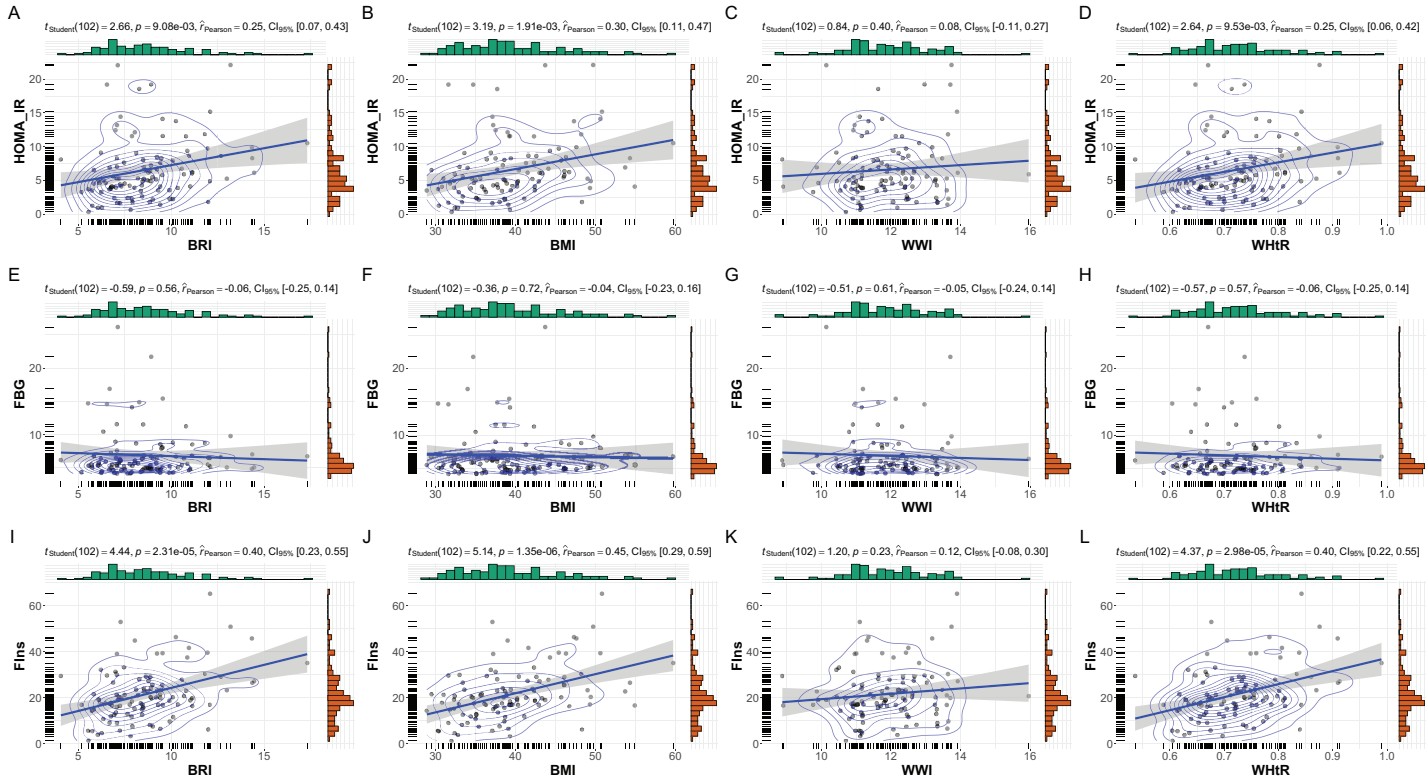

**Figure 2 Correlations between anthropometric indices and metabolic markers in MetS patients.** Correlations between anthropometric indices and metabolic markers in patients with MetS. This figure presents scatterplots and statistical analyses illustrating the relationships between anthropometric indices (BRI, BMI, WHtR, and WWI) and key metabolic markers: (A–D) HOMA-IR, (E–H) FBG, and (I–L) FIns. Correlation coefficients (r), *p* values, and Bayesian inference metrics are annotated within each plot, with shaded regions representing 95% confidence intervals. Significant positive correlations were observed between HOMA-IR and BRI, BMI, and WHtR (*p* < 0.01), as well as between FIns and these indices (*p* < 0.001). Non-significant associations are noted for WWI across all metabolic markers.

## Correlation between anthropometric indices and metabolic markers

The correlations between anthropometric indices (BMI, BRI, WHtR, and WWI) and key metabolic markers (FBG, FIns, and HOMA-IR) are summarized in Fig. 2. Overall, significant positive correlations were observed between most anthropometric indices and metabolic markers, with notable gender-specific variations.

In the overall cohort, BMI showed the strongest correlations with fasting insulin (r = 0.45, *p* < 0.001) and HOMA-IR (r = 0.30, *p* = 0.002). Similarly, BRI and WHtR were positively correlated with fasting insulin (BRI: r = 0.40, *p* < 0.001; WHtR: r = 0.40, *p* < 0.001) and HOMA-IR (BRI: r = 0.25, *p* = 0.009; WHtR: r = 0.25, *p* = 0.010). WWI, however, demonstrated weaker and non-significant correlations with all metabolic markers, including fasting glucose (r = −0.06, *p* = 0.56) and HOMA-IR (r = 0.08, *p* = 0.40).

## Gender-specific subgroup analyses

Gender-specific subgroup analyses revealed notable differences in the relationships between anthropometric indices and metabolic markers, underscoring the influence of

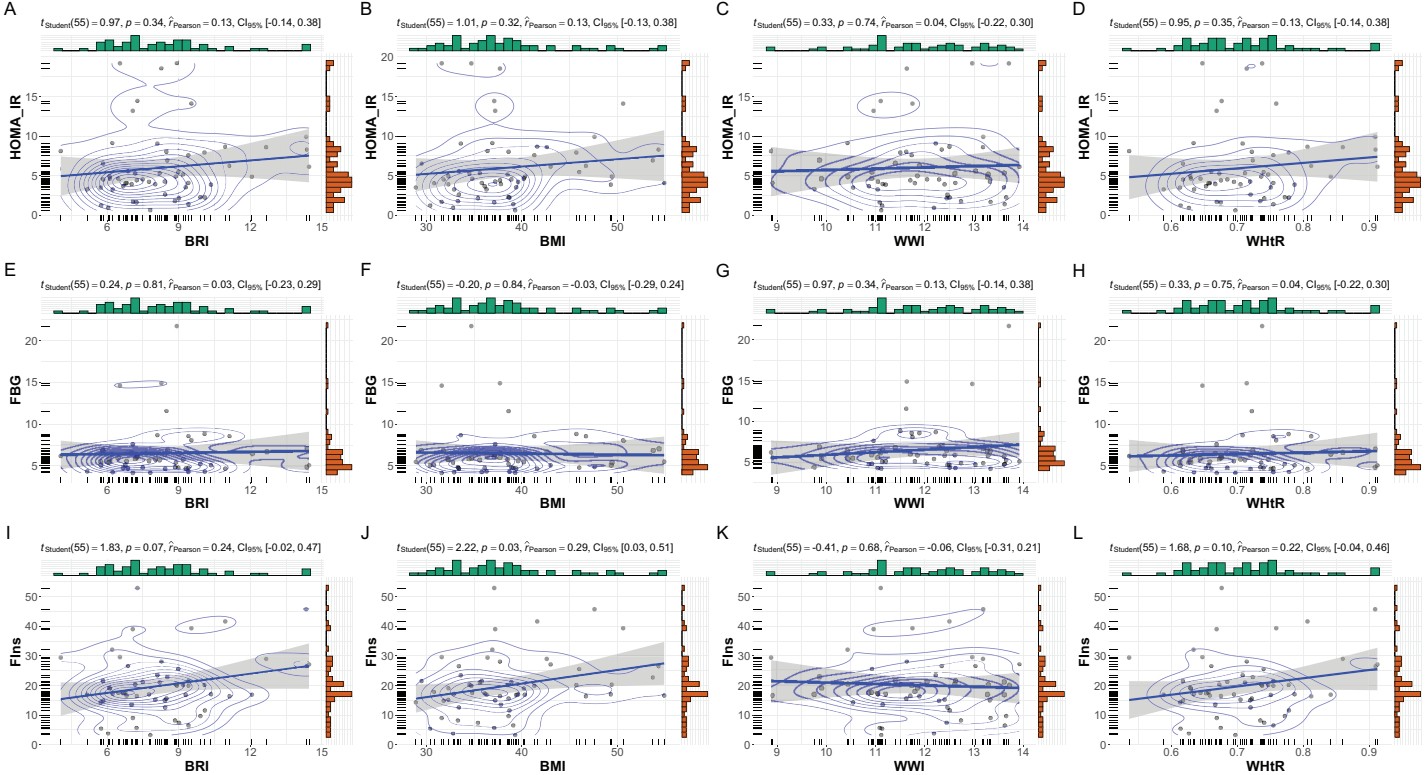

**Figure 3 Subgroup analysis: anthropometric indices *vs.* metabolic markers in female MetS patients.** Subgroup analysis of the relationships between anthropometric indices and metabolic markers in female patients with MetS. This figure depicts scatterplots and statistical analyses examining the correlations between anthropometric indices (BRI, BMI, WHtR, and WWI) and metabolic markers within the female subset (N = 57). (A–D) Correlations with HOMA-IR, (E–H) FBG, and (I–L) FIns are presented. While no significant correlations were observed for HOMA-IR and FBG with these indices, a modest significant correlation was identified between BMI and FIns (p = 0.03, r = 0.29). Bayesian posterior distributions and confidence intervals are illustrated for each analysis, providing robust insight into the strength and uncertainty of the observed relationships.

gender on metabolic profiles in patients with MetS. As shown in Fig. 3, female subgroup patients (*n* = 57) demonstrated weaker and less consistent correlations. BMI showed a moderate but significant association with fasting insulin (r = 0.29, *p* = 0.03), while its correlation with HOMA-IR was not statistically significant (r = 0.13, *p* = 0.32). BRI and WHtR displayed similar patterns, with moderate correlations observed for fasting insulin (BRI: r = 0.24, *p* = 0.07; WHtR: r = 0.22, *p* = 0.10) but non-significant associations with HOMA-IR. WWI exhibited negligible correlations with all metabolic markers in females.

In male subgroup patients (*n* = 47), strong correlations were observed between BMI, BRI, and WHtR with fasting insulin and HOMA-IR (Fig. 4). BMI exhibited the strongest association with fasting insulin (r = 0.60, *p* < 0.001) and HOMA-IR (r = 0.46, *p* = 0.001). Similarly, BRI and WHtR were significantly correlated with fasting insulin (BRI: r = 0.53, *p* < 0.001; WHtR: r = 0.54, *p* < 0.001) and HOMA-IR (BRI: r = 0.35, *p* = 0.01; WHtR: r = 0.36, *p* = 0.01). WWI showed weaker correlations, with no significant associations observed with fasting glucose, fasting insulin, or HOMA-IR.

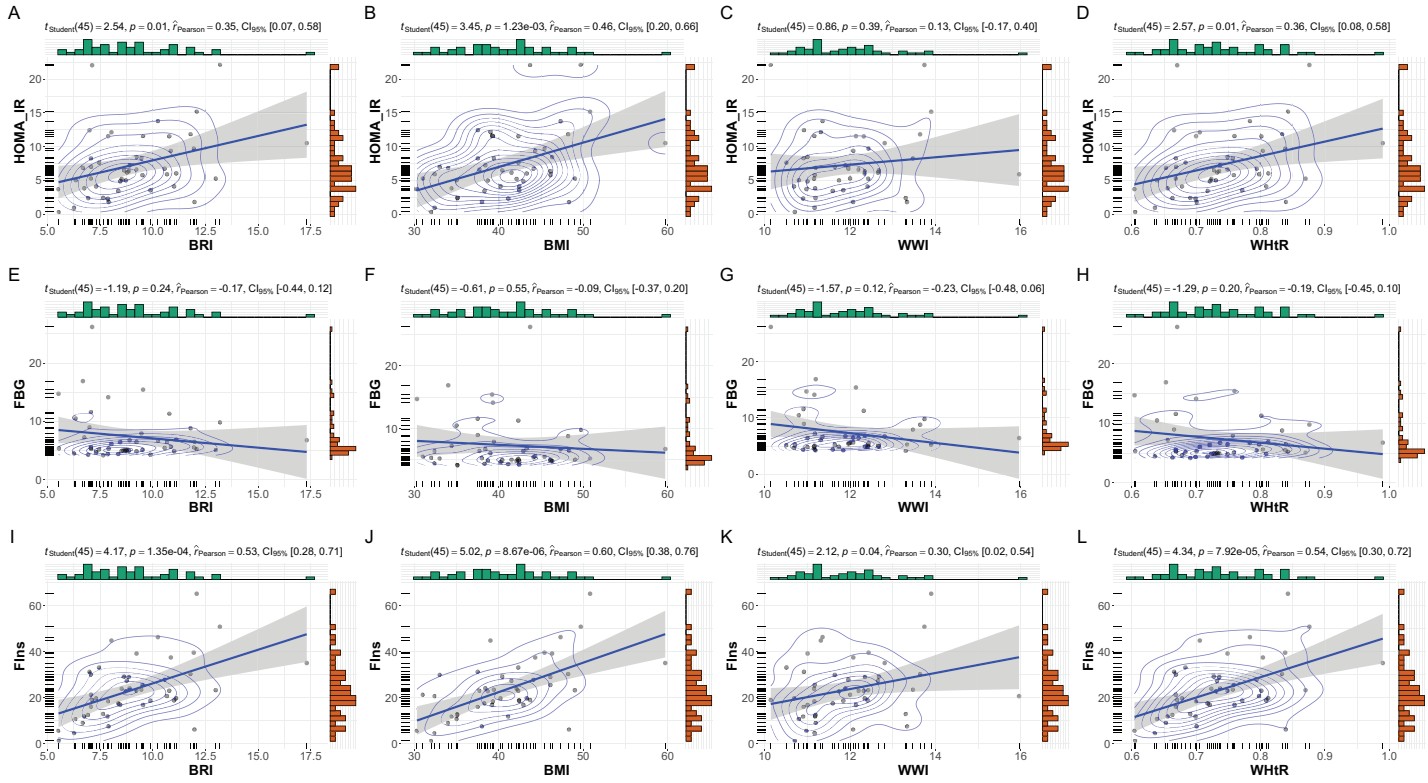

**Figure 4 Subgroup analysis: anthropometric indices *vs.* metabolic markers in male MetS patients.** Subgroup analysis of the relationships between anthropometric indices and metabolic markers in male patients with MetS. This figure illustrates scatterplots and statistical analyses assessing the correlations between anthropometric indices (BRI, BMI, WHtR, and WWI) and metabolic markers within the male subset ($N = 47$). (A–D) Correlations with HOMA-IR, (E–H) FBG, and (I–L) FIns are presented. Significant positive correlations were observed between HOMA-IR and BMI, BRI, and WHtR ($p < 0.05$). Strong correlations were also identified between FIns and BMI, BRI, and WHtR ($p < 0.001$), with moderate significance for FIns and WWI ($p = 0.04$, $r = 0.30$). Bayesian posterior distributions and confidence intervals further highlight the strength and variability of these relationships.

# DISCUSSION

This study highlights significant gender-specific differences in the relationships between anthropometric indices and metabolic markers in patients with MetS. Key findings indicate that BMI, BRI, and WHtR are strongly associated with metabolic dysfunction, particularly in males, while WWI showed limited predictive value across both genders. This observation aligns with the findings (*Geer & Shen, 2009*), who reported higher visceral adiposity and IR burden among women with comparable BMI. Moreover, *Wu et al. (2021)* found that WHtR and BRI were more strongly associated with metabolic risk in women than in men, supporting our gender-stratified correlation results.

To our knowledge, this is among the few studies that simultaneously examines multiple emerging anthropometric indices—including WWI and BRI—in relation to insulin resistance markers in a well-defined MetS cohort, with specific emphasis on gender-stratified patterns. This multifactorial approach offers novel insights into how gender modulates the link between central adiposity and insulin dynamics. Our findings

align with previous research suggesting that men are more prone to visceral fat accumulation, contributing to a stronger association between anthropometric indices and insulin resistance markers such as HOMA-IR (*Ivan et al., 2024*; *Zhang et al., 2024a*). Women, on the other hand, tend to accumulate more subcutaneous fat, which may explain the weaker correlations observed in female patients (*Dahlmann et al., 2023*; *Liu et al., 2024*). This gender disparity in fat distribution underscores the need for tailored approaches when evaluating obesity-related risks in clinical practice (*Wróblewski et al., 2023*; *An et al., 2024*; *Mai et al., 2024*).

The limited utility of WWI in predicting metabolic dysfunction observed in this study contrasts with previous research emphasizing its role in obesity assessment (*Li, Zhao & Wang, 2024*; *Yu et al., 2024*). This discrepancy may arise from differences in study populations, ethnic variations, or the inherent complexity of MetS pathophysiology (*Zierfuss et al., 2020*). These findings underscore the need for further research to validate the relevance of WWI in diverse cohorts and to investigate alternative indices that may offer improved risk stratification (*Cao et al., 2024*).

Consistent with recent literature, our results confirm that BMI and WHtR are reliable indicators of metabolic dysfunction (*Liu et al., 2018*; *Ge et al., 2021*; *Ramesh et al., 2024*). Notably, BRI demonstrated correlations with fasting insulin and HOMA-IR that were comparable to, or stronger than, those of BMI and WHtR. This suggests BRI's potential utility as an alternative marker for central obesity and insulin resistance, particularly in male patients, where these associations were more pronounced. Strengths of our study include the use of standardized laboratory markers (FBG, FIns, HOMA-IR) alongside validated anthropometric indices, and a real-world clinical sample reflective of routine MetS management. Our gender-specific analyses further enhance the clinical interpretability of the results.

While this study benefits from a well-characterized cohort and comprehensive assessments, several limitations should be noted. First, the retrospective design limits causal inferences between anthropometric indices and metabolic dysfunction. Second, the relatively small sample size reduces statistical power and may obscure subtle gender-specific differences, while also increasing the risk of sampling bias. Third, the study's focus on a single population limits generalizability, as genetic, environmental, and cultural factors may influence the predictive value of indices like BMI, WHtR, and BRI in other populations. Additionally, the exclusion of participants with missing data may introduce selection bias, and potential confounders such as dietary intake, physical activity, and socioeconomic factors were not accounted for. Moreover, although dyslipidemia (triglycerides and HDL-C) and blood pressure were included as diagnostic criteria for MetS, they were not analyzed as continuous variables in the correlation models. Consequently, the findings primarily reflect the obesity–insulin resistance axis of MetS rather than its full clinical spectrum. Hormonal status, including menopausal transition, and detailed endocrine profiles were not available in this cohort, which may have influenced the interpretation of gender-related differences in body composition and metabolic risk. Furthermore, direct measures of fat distribution—such as DEXA-derived

visceral and subcutaneous adiposity—were not performed, limiting validation of anthropometric indices against imaging-based metrics.

Future studies should adopt prospective, multi-ethnic designs with larger cohorts to enhance generalizability. Integrating hormonal assessments, imaging-based fat distribution analyses, and lifestyle or behavioral data would provide a more comprehensive understanding of MetS and its gender-specific characteristics. Longitudinal studies are also warranted to clarify temporal relationships between anthropometric indices and metabolic dysfunction, offering deeper insights for prevention and targeted interventions.

## CONCLUSIONS

In conclusion, our results highlight the importance of gender-specific considerations in the assessment of MetS. Anthropometric indices such as BMI, BRI, and WHtR should be prioritized for evaluating metabolic risk, particularly in males, while recognizing the limitations of WWI. These insights may inform targeted interventions and personalized management strategies for MetS patients.

### Funding
This study was funded by the Medical Joint Fund of Jinan University (No.YXZY2024020), and the National Natural Science Foundation of China (No.82402345, No.82371998), and Science and Technology Projects in Guangzhou (No.2024A03J0828). The funders had no role in study design, data collection and analysis, decision to publish, or preparation of the manuscript.

### Grant Disclosures
The following grant information was disclosed by the authors:
Medical Joint Fund of Jinan University: YXZY2024020.
National Natural Science Foundation of China: 82402345, 82371998.
Science and Technology Projects in Guangzhou: 2024A03J0828.

### Competing Interests
The authors declare that they have no competing interests.

### Author Contributions
- Yingxin Li conceived and designed the experiments, performed the experiments, analyzed the data, authored or reviewed drafts of the article, and approved the final draft.
- Huanhua Wu conceived and designed the experiments, performed the experiments, analyzed the data, prepared figures and/or tables, authored or reviewed drafts of the article, and approved the final draft.
- Yingtong Lu performed the experiments, analyzed the data, prepared figures and/or tables, and approved the final draft.
- Jingjie Shang performed the experiments, analyzed the data, prepared figures and/or tables, and approved the final draft.

- Yong Cheng performed the experiments, analyzed the data, prepared figures and/or tables, and approved the final draft.
- Xiaozheng Cao performed the experiments, analyzed the data, prepared figures and/or tables, and approved the final draft.
- Zhiqiang Tan performed the experiments, analyzed the data, prepared figures and/or tables, authored or reviewed drafts of the article, and approved the final draft.
- Qijun Cai performed the experiments, analyzed the data, prepared figures and/or tables, and approved the final draft.
- Hao Xu conceived and designed the experiments, authored or reviewed drafts of the article, and approved the final draft.
- Bin Guo conceived and designed the experiments, authored or reviewed drafts of the article, and approved the final draft.

### Human Ethics

The following information was supplied relating to ethical approvals (*i.e.*, approving body and any reference numbers):

The First Affiliated Hospital of Jinan University approval to carry out the study within its facilities (Ethical Application Ref: KY-2024-184).

### Data Availability

The data is available at Figshare: Guo, Bin (2025). raw_data(renew). figshare. Dataset. https://doi.org/10.6084/m9.figshare.29083457.v1.

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
