# Peer review of "Retrospective study: anthropometric and metabolic characteristics of patients with metabolic syndrome–a gender-specific analysis of clinical and correlation patterns"

_PeerJ, doi:10.7717/peerj.20183_

## Round 0.1 · original submission · Major Revisions

**Language Note:** When you prepare your next revision, please either (i) have a colleague who is proficient in English and familiar with the subject matter review your manuscript, or (ii) contact a professional editing service to review your manuscript. PeerJ can provide language editing services - you can contact us at [email protected] for pricing (be sure to provide your manuscript number and title). – PeerJ Staff

·

Basic reporting

Yingxin Li, Huanhua Wu and et al, have provided a good overview of the introduction to their study. However, this introduction requires more detail on the current anthropometric and metabolic characteristics and how they may differ in other parts of the world. Including the current field's knowledge of sex-specific differences in metabolic markers and disease progression would also help strengthen the justification for your study.

Major comments include:
• Metabolic markers mentioned in the abstract methods include fasting blood glucose, fasting insulin, and HOMA-IR alongside BRI, BMI, WHtR, and WWI. However, these markers only reflect two components of MetS (high fasting glucose/insulin resistance and obesity). It may not be appropriate to group them under the broader category of MetS without including additional markers such as dyslipidemia or hypertension.

Minor comments include:
• Line 84: Keep consistent with the rest of the text, metabolic syndrome has been abbreviated already, no need to write it in full here
• Line 102: Define ‘IR’
• Lines 103 and 104: Make sure there are references here
• Line 141: Spelling mistake for ‘fasting’ blood glucose
• Line 143: Specify here which standardized enzymatic methods or reference to another paper
• Table 1: include the MetS threshold used in this table for clearer characterisation of the patients, do they all express the same 3 comorbidities?
• Figure legends from Figure 1 and Figure 4 have inconsistent line spacing compared to Figure 2 and Figure 3
• Figures 2-4 need to be enlarged; they are too small and blurry, making it difficult to read

Experimental design

• It is great to see that individuals with blood pressure of >130/85 mmHg or those taking antihypertensive medications are included, as they still meet the criteria for hypertension. However, it is equally important to include patients who are taking glucose-lowering medications, as these patients still meet the clinical definition of impaired glucose regulation. Including these patients will allow a more accurate representation of the MetS population.

• Given the broad range of participants, it is critical to consider hormonal status, particularly in a study related to gender-based differences. Post-menopausal individuals often experience a significant shift in body composition, including changes in fat distribution and, in turn, metabolic risk. Not accounting for hormonal status could hide meaningful interpretations of the gender-related findings. Including hormonal status alongside tools like DEXA scanning would also provide validation of fat distribution and the type of fat accumulation. This, in turn, would strengthen the study’s correlations between metabolic markers and body composition.

Validity of the findings

In this retrospective study conducted by Yingxin Li and Huanhua Wu et al, the relationship between anthropometric indices and metabolic markers was explored. This study highlights the distinct gender-specific patterns of these indices and metabolic markers in a certain population. This study provides exciting and important groundwork in shifting anthropometric indices and metabolic markers based on gender, allowing us to better clinically diagnose metabolic syndrome.

Reviewer 2 ·

Basic reporting

Cardiometabolic diseases represent one of the most significant public health challenges in the modern world. Due to the increasing prevalence of obesity, the incidence of metabolic syndrome and its components, such as diabetes, hypertension, and dyslipidemia, is also increasing. All of this contributes to increased cardiovascular risk. Cardiovascular diseases, in turn, are a leading cause of morbidity and mortality in the modern world. Therefore, effective preventive and therapeutic measures are essential. Improving knowledge about the clinical and epidemiological characteristics of individuals with metabolic syndrome is crucial, as this can contribute to improving the quality of care for this patient population. The topic addressed by the authors of this paper is therefore extremely important. In my opinion, the presented manuscript is quite well prepared and demonstrates some substantive and cognitive value. However, I would like to suggest certain changes that, in my opinion, are necessary to further improve the quality and attractiveness of the presented manuscript. I present my suggestions below.

1) In the introduction, the authors drew attention to various parameters derived from basic anthropometric measurements that are helpful in assessing fat tissue distribution and metabolic dysfunction (BMI, BRI, WHtR, WWI). I also suggest mentioning here that bioelectrical impedance body composition analysis is a valuable complement to these measurements, as it provides precise information on the percentage of fat, water, and muscle in a given body mass.

2) The purpose of the work must be precisely defined. You cannot write "to address this gap" in direct reference to the previous part of the text, but simply define the purpose of the work precisely.

3) The exclusion criteria were formulated too broadly and imprecisely. What does "severe medical conditions" mean? Did any surgical intervention ever disqualify participants from participating in the study? The exclusion criteria should be formulated more precisely.

4) The description of the study group lacks information on chronic diseases. What percentage of study participants were treated for diabetes, hypertension, chronic kidney disease, or atherosclerotic diseases?

5) I believe the discussion is too brief and refers to too few references. The results should be carefully compared with those of other researchers. The novelty of the research should be more clearly emphasized. The strengths and limitations of the study should also be carefully discussed.

Experimental design

-

Validity of the findings

-

---

## Round 0.2 · accepted · Accept

Thank you for revising your manuscript to address the reviewers' concerns. Reviewer 1 now recommends acceptance and I am satisfied with the revisions you have made in response to the earlier comments of reviewer 2. The manuscript is now ready for publication.

·

Basic reporting

No comment

Experimental design

No comment

Validity of the findings

No comment

Additional comments

Thank you for addressing the comments.